# Fault Diagnosis of a Multistage Centrifugal Pump Using Explanatory Ratio Linear Discriminant Analysis

**DOI:** 10.3390/s24061830

**Published:** 2024-03-13

**Authors:** Saif Ullah, Zahoor Ahmad, Jong-Myon Kim

**Affiliations:** 1Department of Electrical, Electronic and Computer Engineering, University of Ulsan, Ulsan 44610, Republic of Korea; saifuou@mail.ulsan.ac.kr (S.U.); zahooruou@mail.ulsan.ac.kr (Z.A.); 2Prognosis and Diagnostic Technologies Co., Ulsan 44610, Republic of Korea

**Keywords:** fault diagnosis, linear discriminant analysis, vibration signals, centrifugal pump, K-nearest neighbor

## Abstract

This study introduces an innovative approach for fault diagnosis of a multistage centrifugal pump (MCP) using explanatory ratio (ER) linear discriminant analysis (LDA). Initially, the method addresses the challenge of background noise and interference in vibration signals by identifying a fault-sensitive frequency band (FSFB). From the FSFB, raw hybrid statistical features are extracted in time, frequency, and time–frequency domains, forming a comprehensive feature pool. Recognizing that not all features adequately represent MCP conditions and can reduce classification accuracy, we propose a novel ER-LDA method. ER-LDA evaluates feature importance by calculating the explanatory ratio between interclass distance and intraclass scatteredness, facilitating the selection of discriminative features through LDA. This fusion of ER-based feature assessment and LDA yields the novel ER-LDA technique. The resulting selective feature set is then passed into a k-nearest neighbor (K-NN) algorithm for condition classification, distinguishing between normal, mechanical seal hole, mechanical seal scratch, and impeller defect states of the MCP. The proposed technique surpasses current cutting-edge techniques in fault classification.

## 1. Introduction

Multistage centrifugal pumps (MCPs) transform electrical energy into mechanical energy for industrial operations. These pumps use a series of impellers arranged sequentially to guide fluid through the pumps. A recent examination of 437 malfunctioning MCP found that the industry suffered 6128 h of maintenance-related downtime due to the absence of precise and intelligent fault diagnosis (FD), incurring a cost of $50 million USD [1].

The broad range of applications for MCPs means that faults can result in significant consequences, including wasted energy, downtime in industrial processes, economic setbacks, expensive repairs, and compromised safety for staff. To maintain dependable MCP operation, early fault detection is important and should minimize maintenance and repair expenses. Regular monitoring can be carried out through the collaboration of personnel or by utilizing cost-effective and dependable signal-processing and artificial intelligence (AI) methods [2,3,4]. AI methods for FD consist of preprocessing signals and features as well as tasks related to fault identification and classification [5,6].

There are two primary categories for faults in MCP: mechanical faults (MFs) and fluid flow-related faults. While these faults can be interrelated, MFs occur more frequently. Among MCP faults, 34% are related to issues with a mechanical seal (MS). Besides MS failures, problems with MCP impellers can result in either mechanical faults or a combination of mechanical and liquid flow issues. Additionally, MFs in MCPs can manifest as either soft or hard failures. The latter are easily recognizable but soft failures can cause performance deterioration while allowing the pump to continue functioning. Identifying soft failures in a timely manner is therefore important.

A faulty MS can lead to soft defects in MCPs, including fretting, fluid flushing, and shaft wear. Moreover, an impeller defect (ID) can cause both hydraulic and soft mechanical failures [7]. To minimize maintenance expenses and downtime for MCPs, this study focuses on early fault detection, specifically targeting soft defects arising from a mechanical seal hole (MSH), a mechanical seal scratch (MSS), or an ID.

A variation in the stiffness of a mechanical structure due to a defect causes impulses in the vibration signals (VSs), making it a valuable tool for monitoring the condition of an MCP [8,9]. MFs exert a significant impact on a VS obtained from an MCP. These defects render the VS impulsive and non-stationary, necessitating paying attention to the FD [10,11]. These impulses arise at frequencies in the relevant frequency spectrum but are often overwhelmed by background noise due to their low energy [12]. Bearing-fault harmonics can be differentiated from interference noise using a fault-oriented window series based on a Gaussian mixture model. However, methods that rely on narrowband demodulation often struggle to distinguish between fault impulses and interference noise [13]. Moreover, the statistical characteristics of a defective VS vary over time, making them complex and non-stationary [14,15,16]. Fourier transforms, which are particularly useful for analyzing stationary signals, cannot easily handle such changes. To tackle these challenges, a denoising technique, blind source separation (BSS), was proposed. However, BSS necessitates a baseline signal, posing a limitation for complex VS situations [17]. To proficiently analyze non-stationary signals, innovative signal-processing methods, such as the short time Fourier transform (STFT), have been utilized [18,19].

An STFT uses sample windows of fixed sizes and is commonly applied to time–frequency analysis. Nevertheless, optimizing the frequency resolution may lead to a diminished time resolution and vice versa [20,21]. Time–frequency domain (TFD) transforms offer benefits for complex signals [22]. The TFD wavelet transform (WT) is sensitive to non-stationary defect impulses [23,24,25]. A WT is used to preprocess MCP VSs and extract statistical features in FD [26]. Optimal selection of the mother wavelet is important for WT to avoid oscillation effects, and empirical mode decomposition, an adaptive signal decomposition technique, can overcome the limitations of a WT [27,28]. However, empirical mode decomposition is susceptible to mode mixing and extreme interpolation, enhancing the WT’s attractiveness [22]. Another method that uses the S-transform for preprocessing a VS addresses the shortcomings of both WT and STFT [29,30]. Wavelet coherence analysis is a state-of-the-art method used for FD in MCPs. It generates coherograms and then applies deep-learning techniques [31]. This paper diverges from previous studies that concentrated on a specific frequency range, and this study examines vibration modes across the entire spectrum of a defective MCP. These modes related to defects are filtered out to remove interference from other vibrations. This filtered mode results in the fault-sensitive frequency band (FSFB), which helps us to extract important features for identifying faults.

In intelligent FD, after preprocessing the vibration spectrum, extraction and preprocessing of features are crucial [32]. Extracting discriminant statistical features from the VS across the time domain, frequency domain, and TFD is essential [33,34,35,36]. Deep-learning techniques show considerable potential for extracting and classifying features related to faults [37,38,39]. A data-driven strategy has been introduced for bearing FD, extracting statistical features from raw VS in multiple domains, and a novel deep-learning technique has been proposed based on such features [40]. However, VS from MCP under soft-defect conditions differ due to complex fluid and mechanical interactions, making statistical features extracted from raw MCP VS noisy and inadequate for representing MCP fault-related information. Statistical features from raw VS in the time domain can lack sensitivity to incipient defects and are unsuitable for severe defects. Similarly, frequency-domain statistical features from a raw VS can be noisy due to MCP faults occurring at lower frequencies, which are obscured by noise from microstructural vibrations. To tackle such issues, a new MCP VS preprocessing approach has been proposed that calculates modes for MCP defects in a VS, filters these modes from the MCP vibration spectrum, forming the FSFB, and uses it to extract discriminant statistical features in time, frequency, and TFD.

In previous studies, the emphasis has been on refining VS analysis and statistical feature preprocessing to extract discriminant fault information. Several techniques for feature discrimination evaluation and dimensionality reduction have emerged, with linear discriminant analysis (LDA) and principal component analysis (PCA) among the more prominent [41]. A comparative analysis of dimensionality-reduction methods for MCP FD reportedly produced promising results in PCA [42]. The components generated by PCA capture various symptoms of machinery faults. However, PCA cannot estimate intraclass separability and suffers from information loss. In contrast, LDA aims for an optimized and reduced dimensional representation by taking into account both interclass scatteredness and intraclass separability. As with any supervised learning technique, a large, labeled dataset is required to achieve useful results [43]. Variants such as robust linear optimized LDA [44] and trace ratio LDA [45] have been proposed but face issues due to the penalty graph in intraclass discrimination, undermining classification precision. These limitations emphasize the need for a new explanatory ratio (ER)-LDA technique. The proposed technique begins by evaluating feature informativeness with respect to faults. It calculates the explanatory ratio by comparing the feature’s intraclass degree of scattering to its interclass distance. Features with a high ER undergo LDA to acquire a discriminant set of statistical features with reduced dimensions. The fusion of a feature assessment based on ER and LDA constitutes the novel ER-LDA method. The primary contribution of this work can be summarized as follows:The VS obtained consists of a lot of macrostructural interference noise. To address this concern, the proposed method computes the vibrational modes related to MCP defects and isolates them through filtering, resulting in the formation of the FSFB. This FSFB is important for extracting discriminant statistical features across three domains, which are then integrated into a unified feature pool.The feature pool consists of multiple features. No feature is optimal for accurately defining the MCP condition, and their inclusion could affect the precision of the classification. Therefore, in the proposed method, ER-LDA is used as a technique for extracting discriminant features in MCP FD. ER-LDA evaluates the informativeness of features concerning faults by calculating the explanatory ratio of statistical features. It then applies LDA to highly informative features to achieve reduced-dimension discriminant sets.To ensure practical validation of the approach, the proposed methodology is evaluated using MCP vibration signals obtained from an industrial test rig.

The subsequent sections of this proposed work are structured as follows. Section 2 offers a concise technical background on LDA. Section 3 outlines the setup of the MCP test rig and details the procedure used in the experiment. Section 4 presents and discusses the proposed method. Section 5 presents and elucidates the experimental results. Finally, Section 6 provides concluding remarks and offers potential avenues for future research.

## 2. Technical Background

### Multiclass LDA

LDA is a supervised machine-learning technique for reducing dimensionality by transforming a dataset provided as an input into a lower-dimensional space. It is primarily used in supervised learning scenarios. LDA determines a vector that satisfies two critical properties:It maximizes the sum of the distances between the projections averaged from each class when data points are projected. Essentially, it seeks a vector that maximizes the separation between different classes.It minimizes the variance of projections within the same class during the data-point projection to reduce the variance within each class.

Ultimately, LDA seeks a vector that not only minimizes variance within classes but also maximizes the separation between classes, effectively projecting data points in this optimized space. LDA’s ability to optimize class separation through projection makes it an effective pre-processing step for reducing dimensionality. By maximizing the distinction between different classes of data, LDA creates a lower-dimensional representation that enhances the classification task for subsequent algorithms. This distinct advantage of LDA—amplifying class separability—provides a foundation for algorithms such as k-nearest neighbor (KNN) and support vector machines (SVMs) to operate more effectively and make better-informed decisions in the reduced-feature space [46]. For a training dataset consisting of $x_{1},x_{2},\ldots ,x_{n}$ and two classes $C_{1}$ and $C_{2}$, a mathematical representation of the two critical properties is given in Equation (1). The numerator represents interclass scatter, while the denominator corresponds to intraclass scatter.
(1)$$J\left(X\right)=\frac{{\left(\mu_{1}^{*}-\mu_{2}^{*}\right)}^{2}}{\sigma_{1}^{2}+\sigma_{2}^{2}}$$

In Equation (1), $\mu_{1}^{*}$ and $\mu_{2}^{*}$ represent projected centroids of classes $C_{1}$ and $C_{2}$, respectively, whereas $\sigma_{1}^{2}$ and $\sigma_{2}^{2}$ represent the variances of classes $C_{1}$ and $C_{2}$, respectively. If we have $j^{th}$ number of classes, the projected centroid of the respective $C_{j}$ class will be represented as in Equation (2).
(2)$$\mu_{j}^{*}=X^{T}\mu_{j}$$

Here, X is the projected vector also known as an eigenvector and $\mu_{j}$ is the mean of the respective class. The numerator in Equation (1), which is the distance between the projected centroids of the two classes, can be further simplified as below:(3)$${\left(\mu_{1}^{*}-\mu_{2}^{*}\right)}^{2}={\left[X^{T}\mu_{1}-X^{T}\mu_{2}\right]}^{2}$$
(4)$${\left(\mu_{1}^{*}-\mu_{2}^{*}\right)}^{2}={\left[X^{T}(\mu_{1}-\mu_{2})\right]}^{2}$$
(5)$${\left(\mu_{1}^{*}-\mu_{2}^{*}\right)}^{2}=X^{T}\left(\mu_{1}-\mu_{2}\right){\left(\mu_{1}-\mu_{2}\right)}^{T}X$$
(6)$${\left(\mu_{1}^{*}-\mu_{2}^{*}\right)}^{2}=X^{T}D_{b}X$$

In Equation (6), $D_{b}$ is the scatter between classes. If we have $j^{th}$ number of class, the variance of the respective $C_{j}$ class will be represented as in Equation (7).
(7)$$\sigma_{j}^{2}=\sum_{x_{n}\epsilon C_{j}}\left[\{X^{T}x_{n}-X^{T}\mu_{j}\}{\left\{X^{T}x_{n}-X^{T}\mu_{j}\right\}}^{T}\right]$$
(8)$$\sigma_{j}^{2}=X^{T}[\sum_{x_{n}\epsilon C_{j}}\{x_{n}-\mu_{j}\}{\left\{x_{n}-\mu_{j}\right\}}^{T}]X$$
(9)$$\sigma_{j}^{2}=X^{T}\sigma_{j}X$$

Using Equation (9), the denominator that represents the intraclass scatter can be simplified as given in Equation (10)
(10)$$\sigma_{1}^{2}+\sigma_{2}^{2}=X^{T}\sigma_{1}X+X^{T}\sigma_{2}X$$
(11)$$\sigma_{1}^{2}+\sigma_{2}^{2}=X^{T}\left(\sigma_{1}+\sigma_{2}\right)X$$
(12)$$\sigma_{1}^{2}+\sigma_{2}^{2}=X^{T}D_{w}X$$

In Equation (12), $D_{w}$ is the within-class scatter. We then replaced the values in Equation (1) with Equation (6) and Equation (12) regarding the numerator and denominator, respectively.
(13)$$J\left(X\right)=\frac{X^{T}D_{b}X}{X^{T}D_{w}X}$$

Upon differentiating Equation (13) with respect to X and equating it with zero, it becomes a generalized eigenvalue and eigenvector problem.
(14)$$D_{b}X=\lambda D_{w}X$$

Considering $D_{w}$ is a full rank matrix in Equation (14), we can take its inverse:(15)$$D_{w}^{-1}D_{b}X=\lambda X$$
where λ is the eigenvalue and X is the eigenvector.

## 3. Experimental Test Setup

An MCP setup designed to collect datasets for FD is illustrated in Figure 1. This arrangement comprises a PMT-4008 MCP (Hanil, Gwangju, Republic of Korea) propelled by a 5.5-kW motor; a panel consisting of controllers for temperature, water supply, speed, switch, and flow rate; and a display. The experimental setup involves the utilization of two tanks for water, a main tank and a buffer tank. These tanks were elevated to ensure the maintenance of a net positive suction head at the MCP inlet. The interconnection between the water tanks and the MCP involved steel pipes equipped with pressure gauges and water valves. Figure 2 provides a schematic representation of this experimental setup. 

Once the two tanks were linked to the MCP’s inlet and outlet, the MCP was driven consistently at 1733 rpm. Four accelerometers were used to gather the vibration data from the MCP, positioned strategically with two close to the pump casing and the other two closer to the MS and impellers, each affixed using a sticky substance. All accelerometers functioned independently to record the MCP’s VS. An NI-9234 device was used to digitize the data gathered from the MCP through the accelerometers. For further insight, details regarding the sensors and the digitizing equipment utilized in the procedure are provided in Table 1.

An MS consists of rotating and stationary seals. Two seals with 38 mm inner diameters were used for this study. In the rotating part of both the seals, an intentional hole was drilled, as shown in Figure 3a, keeping the stationary part flawless. The diameter and depth of the hole were both 2.8 mm. These defective seals were used to collect MSH fault data. Figure 3b shows a scratch that was made deliberately on the rotating part of the seal with a length of 10 mm, a diameter of 2.5 mm, and a depth of 2.8 mm, while keeping the stationary part of the seal in a normal condition to gather data for the MSS faults.

Crevice corrosion is a prevalent cause of impeller faults. This study deliberately induced a similar fault with a diameter of 161 mm on a cast-iron impeller by removing a piece of the metal measuring 18 mm in length, 2.5 mm in diameter, and 2.8 mm deep, as shown in Figure 4.

## 4. Proposed Approach

The proposed approach begins with the preprocessing of MCP VS and winds up with the classification of MF in the MCP. The sequence of the proposed FD approach for the MCP is depicted in Figure 5.

Each stage of this proposed approach to FD is explained in this section.

### 4.1. Data Acquisition

At a sampling rate of 25.6 kHz, vibration data from the MCP was gathered for 5 min, resulting in 1200 samples. Among these samples, 300 were obtained during normal operating conditions, another 300 were specifically recorded during an MSH fault, another 300 were acquired in the presence of an MSS fault, and the remaining 300 were collected during the ID condition. These VSs are shown in Figure 6.

### 4.2. Fault-Sensitive Frequency Band Selection

A mechanical defect in the MCP alters the stiffness of the structure, triggering a shock. Generally, such shocks are easily seen in the frequency spectrum at certain points. However, the complexity resulting from the interaction between the mechanical parts and the fluids makes this difficult in MCP. The interaction of the mechanical parts and fluid results in hydraulic faults, and MF cannot be identified by observing the frequency spectrum alone. The negative effect on the statistical features of an unprocessed VS due to macrostructural vibrations is also problematic.

Excitation, generated, and electronic frequencies are the three primary types observed in MCPs. The first two are particularly significant. Along with a thorough understanding of both, familiarization with the frequency harmonics produced by the imbalances in MCPs is equally important. The rotating speed and geometry of MCPs are required parameters for identifying the imbalance fault-sensitive frequencies. For ID, generated frequencies act as a major source for identification. The ID results in an imbalance in the VS. This imbalance can be explained as a fault-sensitive frequency by the following equation:(16)$${ID}_{f}=f_{n}.v$$

In Equation (16), $f_{n}$ represents the frequency harmonics and v is the operating speed of the MCP. Figure 7a depicts the frequency spectrum when the MCP is operating under healthy and normal conditions without facing any defect. Figure 7b depicts the frequency spectrum of the MCP under an ID scenario. The amplitude of ${ID}_{f}$ increased compared with a healthy frequency spectrum at the third, fourth, and fifth harmonics due to the ID.

Fluid interactions with the ID resulted in additional spikes in the ID frequency spectrum. MS defects are related to the excitation frequency of the MCP. The vibration theory of circular rings, based on the potential energy (U) and kinetic energy (K), is used to calculate the excitation frequency. Using the energy conservation principle,
(17)$$\frac{d}{dt}\left(K+U\right)=0$$
(18)$$K=2\pi r{\left(s_{r}^{\prime }\right)}^{2}\left(\frac{\rho A}{2}\right)$$
(19)$$U=\frac{AE\pi s_{r}^{2}}{r}$$

In Equations (18) and (19), r is the ring radius, $s_{r}$ is the radial displacement, and A and E denote the cross-sectional area and elastic modulus, respectively. $\frac{s_{r}}{r}$ in the potential energy equation is the mathematical representation for ring-unit elongation. Simplifying Equation (17) produces Equation (20):(20)$$s_{r}^{\prime \prime }+\omega^{2}s_{r}=0$$
where ω is the angular frequency. The ring fundamental frequency (RFF) can be obtained by solving Equation (20):(21)$$RFF=\sqrt{\frac{E}{\rho }}\left(\frac{1}{2\pi r}\right)$$

Using appropriate modes of vibration, the in- and out-of-plane vibration modes in MS are expressed in Equations (22) and (23), respectively:(22)$$F_{IP}=\left(\frac{2m^{3}-2m}{\pi d^{2}}\right)\sqrt{\frac{Eah^{2}}{12a\rho m^{2}+2th^{3}(1+\nu )}}$$
(23)$$F_{OP}=\left(\frac{m^{3}-m}{\pi \sqrt{m^{2}+1}}\right)\sqrt{\frac{Et^{2}}{3d^{4}\rho }}$$

In Equations (22) and (23), m is the vibration mode, ν is Poisson’s ratio, and a denotes the constant of torsion, while d, t, and h are the diameter, thickness, and height, respectively. In-plane vibrations generally occur at high frequencies, while in the context of an MS, lower frequencies manifest in out-of-plane bending modes of vibrations. Figure 8a–c reveal that an MCP excitation frequency emerges between the second and third modes of flexural vibration, with an amplitude almost twice as large as the normal condition (NC) whenever an MS defect occurs.

To extract distinctive statistical features from a VS and to cater to macrostructural vibrations, this study focuses on the MCP VS up to the third mode of flexural vibration. This involves applying a lowpass filter with a cutoff frequency of 4.6 kHz, which effectively encompasses the fault-sensitive frequencies associated with an ID, corresponding hydraulic defects, and CP excitation frequencies. The filtered modes of vibration result in an FSFB used for statistical feature extraction in the three domains in the next step.

### 4.3. Feature Pool

A combination of features extracted in three domains resulted in the feature pool. A variation in the stiffness of a mechanical structure due to a defect causes impulses in the vibration signals (VSs). These variations in the stiffness result in the change in amplitudes of the signals in the time domain. Therefore, the extraction of statistical features from the time-domain signal can be a valuable tool for monitoring the condition of the MCP. Different statistical features in the time domain are given in Table 2. In time-domain statistical features, $X_{m},X_{r},X_{rms}$, and $X_{p}$ represent the mean, root amplitude, root mean square, and peak value of the signal. Generally, mechanical vibrations are excited by the occurrence of any fault and hence uplift the values of these features. However, these features are not sensitive to weak or emerging faults. Therefore, kurtosis ($X_{kur}$), crest factor ($X_{crest}$), clearance factor ($X_{clearance}$), and impulse factor ($X_{impulse}$) can indicate the existence of the impulse in the vibration signals and are better indicators for incipient faults. $X_{impulse}$ and $X_{clearance}$ are also helpful in indicating the sharp impulses resulting from the mating of the surfaces of the defective part and the bearing. $X_{kur}$ is very sensitive to the faults in the early stage; however, it is not a good indicator for more severe faults as its value starts decreasing at a certain stage of severity instead of increasing further. Hence, it can be seen that the different statistical features in the time domain given in Table 2 have their own importance depending upon the severity of the fault and are equally important in diagnosing any fault. These features compensate each other, and each feature consists of fault information as per its own properties. Therefore, more features are extracted to accurately diagnose machinery faults [47].

Table 3 presents the frequency-domain statistical features used in the study. Mean frequency ($X_{mf}$) indicates the energy of the vibration signal in the frequency domain. As the fault increases, $X_{mf}$ increases accordingly. $\sigma_{f}$ and $X_{rvf}$ represent the standard deviation and root variance frequency, which shows the convergence of the spectrum power. Root mean square frequency ($X_{frms}$) and spectral kurtosis ($X_{spkur}$) indicate the change in the position of the dominant frequencies in the frequency spectrum. Hence, these statistical features in the frequency domain can analyze the fault from different aspects. Therefore, these features are equally important and are selected to achieve more accurate and precise diagnosis results [47].

To obtain the statistical features in the TFD, the FSFB transforms the TFD using a wavelet packet transform (WPT). In this investigation, a Daubechies family db4 mother wavelet was chosen to decompose the FSFB up to m = 3 levels, resulting in a total of $2^{m}$ WPT bases. The features outlined in Table 2 were then extracted from each WPT base. Experimental studies on wavelet selection indicate that Daubechies family db4 mother wavelets are sensitive to processes that occur within an MCP. In the proposed approach, a Daubechies family db4 mother wavelet was chosen for FSFB decomposition. All the extracted statistical features were amalgamated into a combined feature pool, encompassing features from all domains. The feature pool comprises 113 features for every MCP condition.

### 4.4. Explanatory Ratio LDA

The feature pool consists of multiple features. No feature is optimal for accurately defining the MCP condition, and their inclusion could affect the precision of the classification. ER-LDA was introduced to extract the discriminant feature in MCP FD. ER-LDA evaluates feature informativeness regarding faults by calculating the ER of the statistical feature, followed by the application of LDA to highly informative features to achieve reduced-dimension discriminant sets. A step-by-step explanation is given below:

Step 1. Equation (24) is used to calculate the sparseness of the feature between classes denoted by $S_{i}$.
(24)$$S_{i}=\frac{1}{K}\sum_{k=1}^{K}S_{k,i}$$
$S_{k,i}$ can be obtained using Equation (25)
(25)$$S_{k,i}=\frac{1}{N_{k}(N_{k}-1)}\sum_{l,m=1}^{N_{k}}\left|y_{m,k,i}-y_{l,k,i}\right|$$
where $l,m=1,2\ldots N_{k},l\neq m$.

In Equations (24) and (25), *k* denotes the classes, y is the feature, i represents the number of features, and $N_{k}$ is the sample number.

Step 2. The feature mean between classes, $\mu_{k,i}$, is calculated.

Step 3. Equation (26) is used to calculate the distance between the separate classes’ features.
(26)$$D_{i}=\frac{1}{K(K-1)}\sum_{r,s=1}^{R}\left|\mu_{s,i}-\mu_{r,i}\right|$$
where r,s=1,2…R,r≠s.

Step 4. The explanatory ratio of the feature is calculated in this step by using Equation (27), which depends on the values obtained in step 1 and step 3.
(27)$$ER=\frac{D_{i}}{S_{i}}$$

Step 5. Those features with an explanatory ratio greater than 0.5 will be considered helpful while others will be considered less helpful. LDA will be applied to the helpful features of the feature pool.
(28)$$HFP=\left\{\begin{array}{c} ER>0.5,\;helpful\\ otherwise\;less\;helpful \end{array}\right.$$

A set of discriminant features, characterized by high interclass distance, reduced intraclass sparseness, and lower dimensions, was successfully obtained. This new approach, termed ER-LDA, effectively addresses the issues of interclass feature distance and intraclass feature sparseness of traditional LDAs.

### 4.5. K-NN

The resulting features were then utilized in MCP condition classification using the K-Nearest Neighbors (K-NN) algorithm, with K set to three. The choice of K-NN for MCP fault classification is motivated by its low computational cost and straightforward architecture. Tuning of a single hyperparameter K is required in K-NN which makes tuning easy. Another advantage of K-NN is if we add new data to the dataset, there is no need to retrain a new model as the prediction is adjusted and constantly changes with new data itself. 

## 5. Results

The evaluation dataset consists of 1200 VS sourced from the MCP, encompassing conditions such as NC, MSH, MSS, and ID. The feature pool aggregates the extracted statistical feature from these VS values, calculated as N × SFN × VS_ins_. Here, N represents the classes, VS_ins_ signifies the VS instances, and SFN denotes the statistical feature derived from the VS for each class. For validation, a cross-validation approach employing n folds (n = 3) was used. The dataset was divided into three folds, where one fold underwent testing for classifier assessment while the remaining two folds were utilized for training. Among the 1200 samples, 800 were used to train the classifier, and the remaining 400 were dedicated to testing the classifier. Random sample selection was ensured across trials, conducted 25 times to ensure stable results in the classification process.

Following VS and statistical feature preprocessing with the proposed approach, novel fault-related features were extracted. This study extensively evaluated the extracted features for FD and compared them against other methodologies: a time-domain feature extraction method (WPT-BE-MSVM), an unsupervised feature preprocessing technique (PCA) [42], and a supervised feature preprocessing method (Tr-LDA) [41]. The comparison employs evaluation metrics, namely the true positive rate (TPR) or Recall, the average accuracy of classification (AAC) or Precision, and the error rate ($E_{R}$) for classification. These metrics were determined using the following equations: (29)$$TPR\%=\frac{1}{j}\sum_{i=1}^{j}\left[\frac{{TP}^{i,l}}{{TP}^{i,l}+{FN}^{i,l}}\right]\times 100\%$$
(30)$$AAC\%=\frac{1}{j}\sum_{i=1}^{j}\left[\sum_{l=1}^{m}\frac{{TP}^{i,l}}{N}\right]\times 100\%$$
(31)$$E_{R}\%=\frac{1}{j}\sum_{i=1}^{j}\left[\frac{{TP}^{i,l}+{FN}^{i,l}}{{TP}^{i,l}+{FN}^{i,l}+{TN}^{i,l}+{FP}^{i,l}}\right]\times 100\%$$

These equations describe the elements involved in evaluating the performance metrics within the cross-validation framework. N denotes the number of samples, j signifies the fold, i is used for the number of iterations, and ${TP}^{i,l}$, ${FN}^{i,l}$, ${TN}^{i,l}$, and ${FP}^{i,l}$ indicate the true positive, false negative, true negative, and false positive, respectively, in the testing subset used to evaluate the CV process. These elements are utilized in Equations (29)–(31) to compute various performance metrics pertaining to accuracy, precision, and $E_{R}$ across the different conditions or classes. 

The findings presented in Figure 9 and Figure 10 and Table 4 show that, compared with state-of-the-art methods, the proposed method achieved superior performance in identifying MCP working conditions, reaching 100% AAC, 100% TPR, and 0% $E_{R}$. This performance superiority aligns with the method’s approach, which is initiated by computing vibration modes specific to MCP defects.

By filtering out interference from macrostructural vibration noise and isolating MCP defect modes from the overall vibration spectrum, the proposed method constructed the FSFB, which was then used to extract statistical features in multiple domains and combine them into a higher-dimensional feature pool. Recognizing potential noise among these features, the proposed method employed ER-LDA to reduce the dimensionality while extracting discriminant features. ER-LDA evaluated feature relevance regarding faults, after which LDA was applied to features displaying high informativeness. The discriminant feature space resulting from the proposed approach (Figure 9) exhibited significant discriminative capabilities, demonstrating reduced sparsity among features within the same class. Figure 10 confirms the higher AAC achieved by the proposed method. These observations indicate the method’s efficacy in extracting relevant, discriminative features and contributing to superior classification accuracy.

The WPT+BE-MSVM method employs a continuous decomposition of VS into distinct frequency bands using the WPT. This process rearranges WPT nodes based on their energy levels and subsequently extracts significant time-domain features. From each of these nodes, features are extracted and classified utilizing SVMs. The suggested node for MCP FD is determined by the node producing the greatest classification accuracy. In this case, WPT+BE-MSVM achieved an overall AAC of 96.3%, a TPR of 96.2%, and an $E_{R}$ of 7%.

The supervised feature pre-processing technique, Tr-LDA, is a linear dimensionality-reduction method, using trace ratio criteria to increase interclass distances and minimize intraclass dispersion. When applied to our dataset, this algorithm yielded an $E_{R}$ of 16.8%, an AAC of 89.6%, and a TPR of 89.6%. These results indicate underperformance relative to our proposed method. This underperformance aligns with expectations as Tr-LDA transforms data without emphasizing feature processing for the extraction of intrinsic discriminant information from raw statistical features. Although proficient in minimizing intraclass dispersion, Tr-LDA struggled to achieve effective separation among different classes, as evidenced by the 16.8% $E_{R}$ observed (Figure 10).

PCA operates as an unsupervised method for feature processing, reducing dimensionality by capturing data variance and creating a condensed data representation. When used with a KNN classifier on our dataset, PCA yielded an $E_{R}$ of 23.5%, an AAC of 83%, and a TPR of 82%. These results, shown in Table 4 and Figure 10, indicate its poor performance relative to the proposed method. Determining the optimal number of components in PCA to retain important fault-related data features presents a challenge. Figure 9 underscores this difficulty, emphasizing that the feature classification achieved by PCA is not as robust as that of the proposed technique. This disparity in feature classification quality contributes to the higher classification error observed in PCA, emphasizing its limitations while comparing it to the efficacy of the proposed approach.

## 6. Conclusions

The paper introduces a novel method for condition classification in MCP. In the signal pre-processing phase, this method identifies an FSFB from raw VS data. This selection involves calculating vibration modes specific to pump defects, subsequently filtering these modes to isolate FSFB. From these bands, statistical features across time, frequency, and time–frequency domains are extracted and consolidated into a unified feature pool. To reduce dimensions while extracting discriminant features from this pool, a novel ER-LDA approach is employed. ER-LDA evaluates feature relevance concerning faults by assessing explanatory ratios, using LDA to retain features with high informativeness. The MCP condition classification of the resulting discriminant feature set is carried out using K-NN. The proposed technique demonstrates superior results compared with existing state-of-the-art methods, achieving an AAC of 100%. However, the highly separable and compact feature space generated by this method may pose challenges if different classification algorithms other than K-NN are employed. Future endeavors should apply the proposed method on a varying-speed dataset of MCP. An additional recommendation for future work is the application of this method to fluid-related defects in MCP.

## Figures and Tables

**Figure 1 sensors-24-01830-f001:**
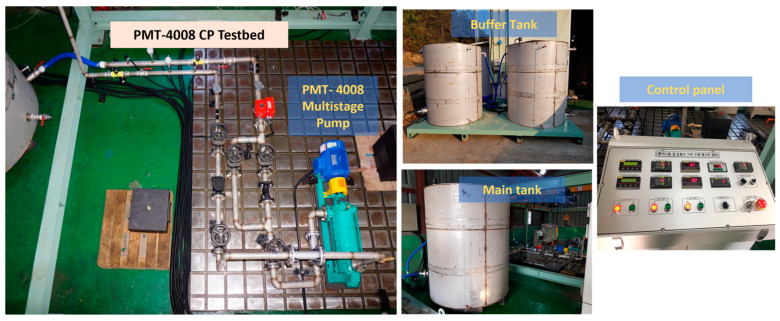
MCP test setup for data collection for fault diagnosis.

**Figure 2 sensors-24-01830-f002:**
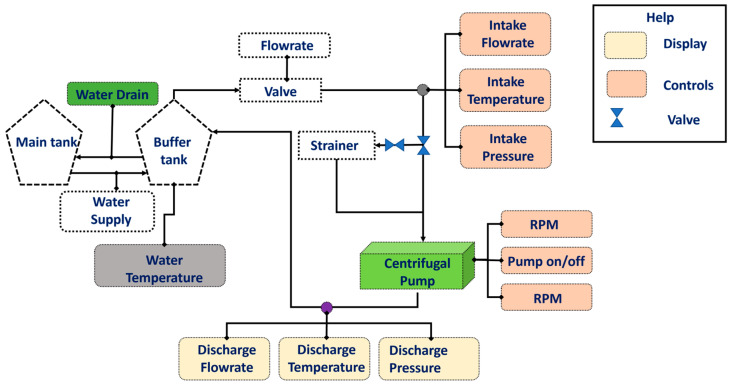
MCP test setup schematics.

**Figure 3 sensors-24-01830-f003:**
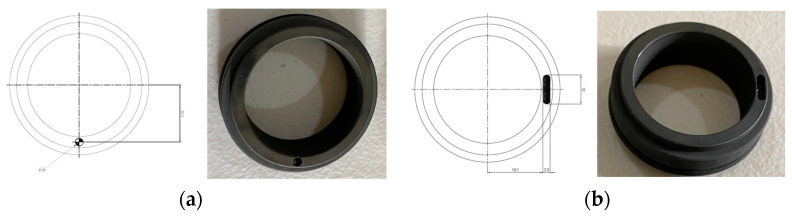
(**a**) Mechanical seal hole and (**b**) a mechanical seal scratch.

**Figure 4 sensors-24-01830-f004:**
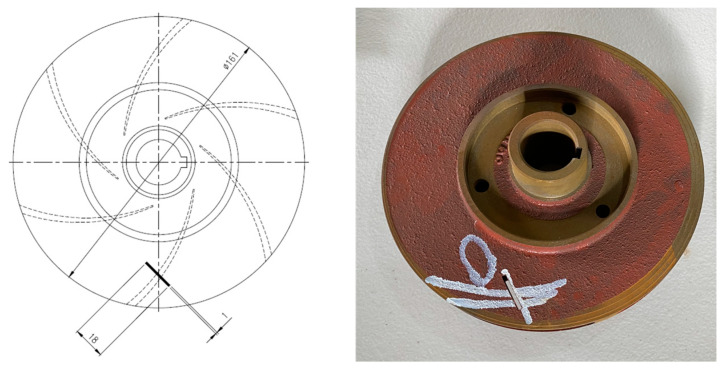
Impeller defect.

**Figure 5 sensors-24-01830-f005:**
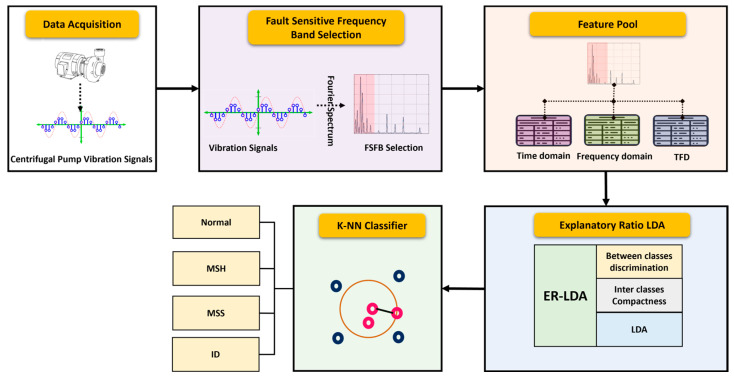
Sequential diagram of proposed FD approach of MCP.

**Figure 6 sensors-24-01830-f006:**
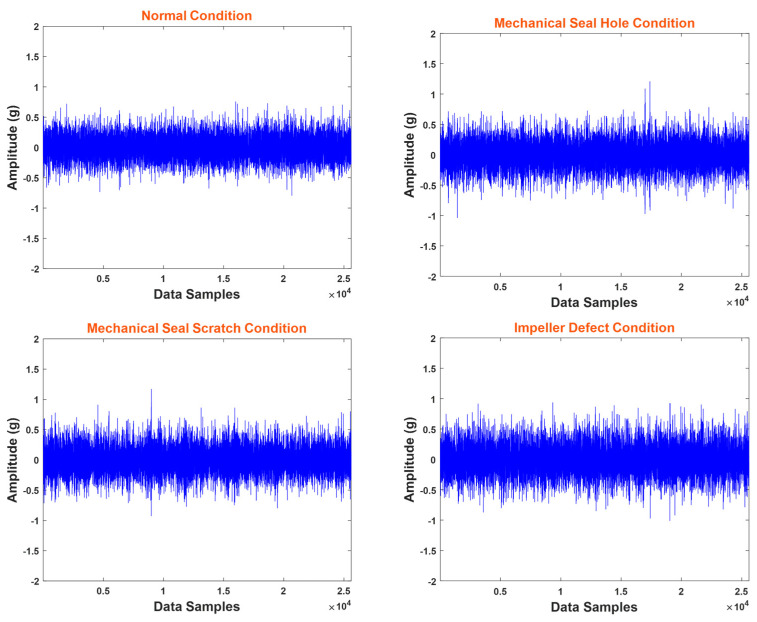
Vibration signals under different conditions of MCP.

**Figure 7 sensors-24-01830-f007:**
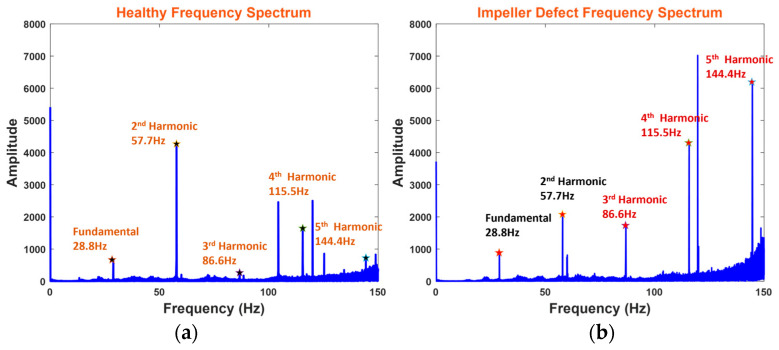
MCP frequency spectrum: (**a**) healthy condition and (**b**) impeller defect condition.

**Figure 8 sensors-24-01830-f008:**
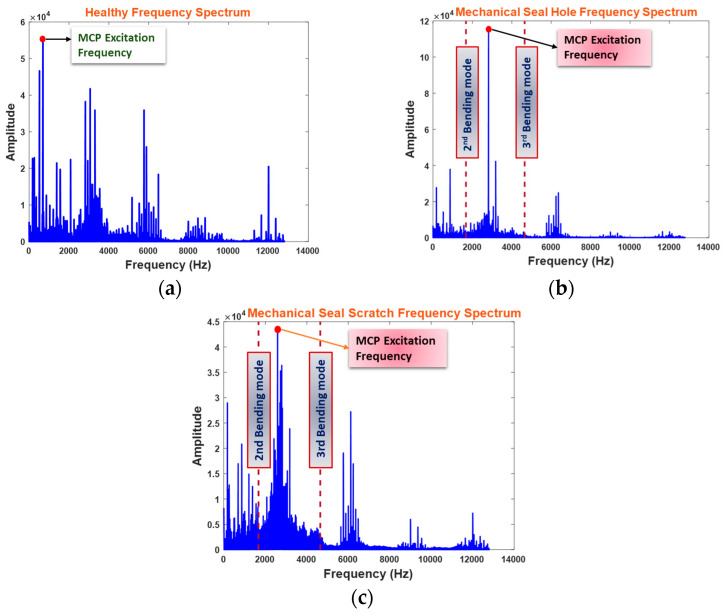
MCP frequency spectrum: (**a**) Healthy condition, (**b**) MSS condition, and (**c**) MSH condition.

**Figure 9 sensors-24-01830-f009:**
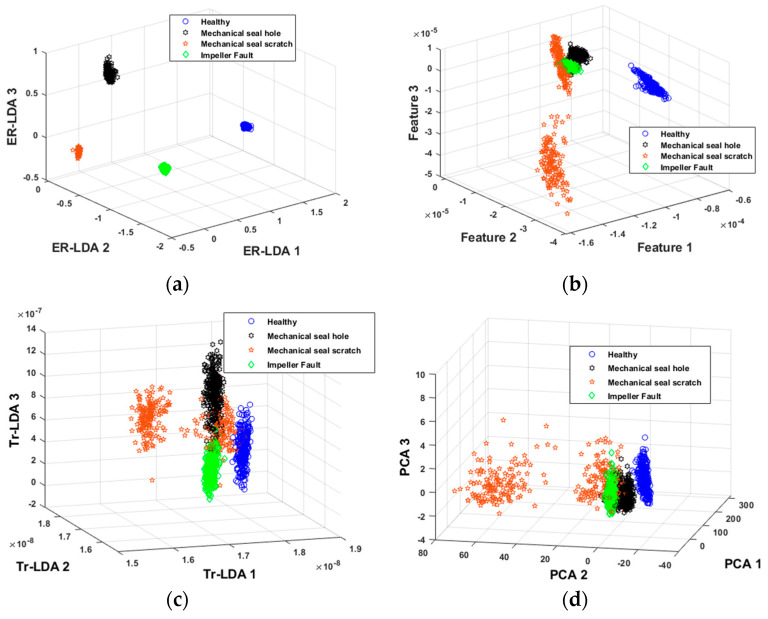
3D presentation of feature pool: (**a**) proposed method, (**b**) WPT-BE-MSVM, (**c**) Tr-LDA, and (**d**) PCA.

**Figure 10 sensors-24-01830-f010:**
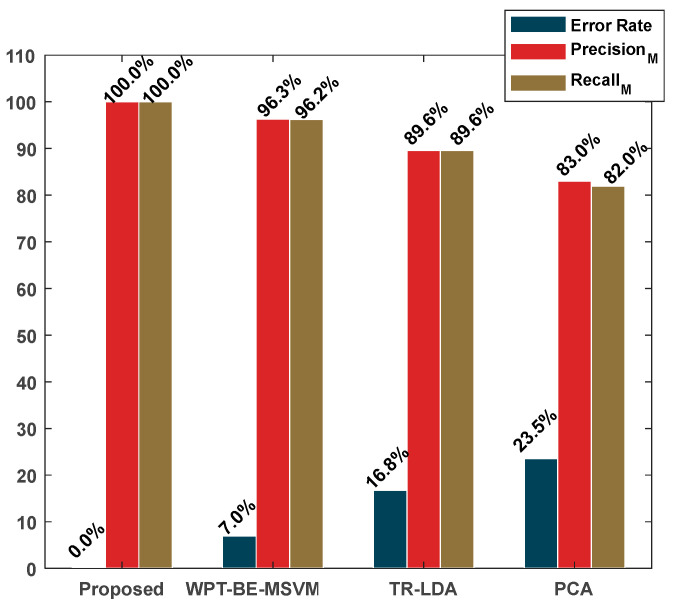
Error rate, precision, and recall obtained from the proposed and reference methods.

**Table 1 sensors-24-01830-t001:** Instrument details for data collection.

Instrument Name	Specification	Value
Accelerometer 622b01	Sensitivity	100 mV/g (10.2 mV/(m/s^2^)) ± 5%
	Frequency range	0.42–10 kHz
DAQ NI-9234	Generator	24 bits ADC resolution and 4 analog input channels
	Frequency range	0–13.1 MHz

**Table 2 sensors-24-01830-t002:** Time-domain statistical features.

Statistical Feature	Formula	Statistical Features	Formula
RMS	$X_{rms}=\sqrt{\frac{\sum_{n=1}^{N}{\left[x(n)\right]}^{2}}{N}}$	Kurtosis	$X_{kur}=\frac{\sum_{n=1}^{N}{\left[x\left(n\right)-X_{m}\right]}^{4}}{(N-1)X_{sd}^{4}}$
Shape factor	$X_{sf}=\frac{X_{rms}}{\frac{1}{N}\sum_{n=1}^{N}\left|x\left(n\right)\right|}$	Variance	$X_{v}=\frac{\sum_{n=1}^{N}{\left[x\left(n\right)-X_{m}\right]}^{2}}{\left(N-1\right)}$
Root amplitude	$X_{r}={\left[\frac{\sum_{n=1}^{N}\sqrt{\left|x\left(n\right)\right|}}{N}\right]}^{2}$	Peak value	$X_{p}=max|x\left(n\right)|$
Mean	$X_{m}=\frac{\sum_{n=1}^{N}x(n)}{N}$	Crest factor	$X_{crest}=\frac{X_{p}}{X_{m}}$
Standard deviation	$X_{sd}=\sqrt{\frac{\sum_{n=1}^{N}{\left[x\left(n\right)-X_{m}\right]}^{2}}{\left(N-1\right)}}$	Clearance factor	$X_{clearance}=\frac{X_{p}}{X_{root}}$
Skewness	$X_{skew}=\frac{\sum_{n=1}^{N}{\left[x\left(n\right)-X_{m}\right]}^{2}}{(N-1)X_{sd}^{3}}$	Impulse factor	$X_{impulse}=\frac{X_{p}}{\frac{1}{N}\sum_{n=1}^{N}\left|x\left(n\right)\right|}$

**Table 3 sensors-24-01830-t003:** Frequency-domain statistical features.

Statistical Feature	Formula	Statistical Features	Formula
Mean frequency	$X_{mf}=\frac{\sum_{k=1}^{K}s(k)}{K}$	Spectral kurtosis	$X_{spkur}=\frac{\sum_{k=1}^{K}{\left[s\left(k\right)-X_{mf}\right]}^{4}}{(K-1)\sigma_{f}^{4}}$
Standard deviation	$\sigma_{f}=\sqrt{\frac{\sum_{k=1}^{K}{\left[s\left(k\right)-X_{mf}\right]}^{2}}{\left(K-1\right)}}$	Root mean square frequency	$X_{frms}=\sqrt{\frac{\sum_{k=1}^{K}f_{k}^{2}(s(k){)}^{2}}{s(k)}}$
Root variance frequency	$X_{rvf}=\sqrt{\frac{\sum_{k=1}^{K}{\left(s\left(k\right)-X_{mf}\right)}^{2}}{K}}$		

**Table 4 sensors-24-01830-t004:** Results obtained from proposed and state-of-the-art methods.

	Healthy	MSH	MSS	ID	ACA
Proposed	100	100	100	100	100%
WPT-BE-MSVM	100	96.50	97.82	90.49	96.20%
Tr-LDA	87.73	94.23	76.82	100	89.56%
PCA	97.70	80.35	64.04	85.71	81.95%

## Data Availability

The data are from the industry. They are not available publicly due to the privacy policy of the industry.

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
