# Peer review of "Fault Diagnosis of a Multistage Centrifugal Pump Using Explanatory Ratio Linear Discriminant Analysis"

_sensors, 2024, doi:10.3390/s24061830_

Round 1

Reviewer 1 Report

Comments and Suggestions for Authors

Abbreviations are always singular, i.e., MCP is fine for both singular and plural cases. Do not use MCPs. The same may be observed for the usage of other abbreviations such as MF, VS etc. Please do not pluralize these abbreviations by using MFs and VSs. Moreover, the authors are using abbreviations way too much. For example, you don't need to abbreviate the term frequency spectrum and call it FS. The readability and understandability of the manuscript suffers a lot. Use abbreviations only when necessary. Do not go for the overkill.

Lines 84 to 89 in the introduction section where the authors differentiate their approach from existing approaches can be further improved for clarity.

The authors are encouraged to further improve the introduction section by referring to similar work in relevant domains.

The authors are encouraged to revise the statement made regarding the contribution of their work by citing the problem or sub-problem that their work solves or addresses.

In almost all the equations, especially in the Section on LDA, the authors need to specify what quantity does each symbol denote. The authors may also consider reducing the number of equations here and cite only those which are necessary in understanding their work.

Section 3 can be restricted only to the description of the experimental test rig and the dataset collected and used for experimentation, whereas the procedure, which sounds like something to do with the proposed methodology can be merged in Section 4, where the authors talk about their proposed methodology. Section 2, where the authors provide technical background on LDA can be removed. A brief description of LDA can be provided as part of Section 4 on their proposed methodology. This reorganization will change the number and numbering of Sections.

Please refer to line 365---Is there any special reason for choosing an ER threshold of 0.5?

Comments on the Quality of English Language

The quality of English is reasonable but can be improved further. The authors are requested to revise their usage of abbreviations, which I guess, is more than required and affects the readability of their work, i.e., readers are forced to look up an abbreviation / acronym at every nook and corner of the paper.

Reviewer 2 Report

Comments and Suggestions for Authors

This paper presents an approach to fault diagnosis of a multistage centrifugal pump (MCP) using explanatory ratio (ER) linear discriminant analysis (LDA). It is a well-structured paper with interesting results. However, it requires further improvements before publication.

1. The abstract should be rewritten to reflect the significance of the proposed work. The current abstract shows a lot of background information.

2. The main contributions of this paper should be further summarized and clearly demonstrated.

3. The method in the context of the proposed work should be written in detail.

4. In Eq.(1)~Eq.(14), the physical meanings of variables, parameters, constants, etc. need to be provided to facilitate the author's understanding.

5. In your study,the MCP was driven consistently at 1733 rpm, does speed of  MCP affect your experimental results and conclusions?

6. Figure 10 is insufficient clarity, please revise it to be clear.

7. Some new references should be added to improve the reviews the literatures. For example, https://doi.org/10.1109/JSEN.2022.3179165; https://doi.org/10.1016/j.ijnaoe.2023.100557 and https://doi.org/10.1109/TIM.2022.3159005 etc.

8. There are a few typos and grammar errors in the manuscript. Please polish the manuscript carefully.

Comments on the Quality of English Language

 Moderate editing of English language required

Reviewer 3 Report

Comments and Suggestions for Authors

This paper proposes a method for fault diagnosis in a multistage centrifugal pump using Explanatory Ratio (ER) Linear Discriminant Analysis (LDA). The proposed method calculated the feature importance based on the ER between interclass distance and intraclass scatteredness, applying LDA to features with high ER to generate a selective feature set. This set is then classified using a k-nearest neighbour (KNN) algorithm. To further improve this work, some suggestions are given as follows:

1.     In this paper, the KNN algorithm is used to condition classification. The article does not mention why the KNN algorithm was chosen over other algorithms. Compared to other algorithms, what are the advantages of KNN, and why did the author choose KNN? Please provide a detailed explanation.

2.     Elaborate on the process for selecting high ER features and the criteria used.

3.     Section 2.1 contains many formulas, but the article does not provide detailed explanations for each formula. Meanwhile, some sentences in this part are confusing. Please carefully review and correct the issues.

4.     The article utilizes 17 types of features from time and frequency domains. How were these features chosen? Is every feature useful?

5.     Please check for grammatical errors, typos, spelling errors, and formatting inconsistencies and correct them in the manuscript.

6. Please consider to enhance the lit review by including more intelligent diagnosis methods, such as time series methods (10.1109/TIM.2023.3259048;10.1016/j.ymssp.2022.108907), and trustworthy machine learning (10.1016/j.ins.2023.119496, 10.1016/j.energy.2023.127033).

Comments on the Quality of English Language

NA

Reviewer 4 Report

Comments and Suggestions for Authors

In this paper, a fault diagnosis method of multistage centrifugal pump is proposed.The content of the paper has certain research value. The manuscript requires some improvement.

1. All formula symbols need to be interpreted

2. The sensor installation mode needs to be described

3. The classification effect is inconsistent with different model parameters, and the parameters of the studied identification strategy need to be explained

4. The composition of the dataset requires further explanation

5. The data processing method needs to be explained. Does it need to be dimensionless?

6. The font in the picture is too small to read

7. The expression in Figure 6 is not very clear and does not explain the mentioned method. The number of features and the correlation of features need to be analyzed

8. Are the parameters of classification algorithm KNN consistent?

Comments on the Quality of English Language

Minor editing of English language required

Round 2

Reviewer 2 Report

Comments and Suggestions for Authors

I have appreciated the deep revision of the contents and the present form of this manuscript. All my previous concerns have been accurately addressed. I think that this paper can be accepted.

Comments on the Quality of English Language

Minor editing of English language required

Reviewer 4 Report

Comments and Suggestions for Authors

The questions raised have been revised, and it is suggested to add some latest research results in Chapter 1, and explain the comparative advantages of the methods of this paper.

Comments on the Quality of English Language

Minor editing of English language required, It is recommended to read the whole text and correct some grammatical errors.